# Biosynthetic Pathways of Tryptophan Metabolites in *Saccharomyces cerevisiae* Strain: Insights and Implications

**DOI:** 10.3390/ijms25094747

**Published:** 2024-04-26

**Authors:** Hsin-Chieh Kung, Ngoc-Han Bui, Bo-Wun Huang, Nicholas Kiprotich Cheruiyot, Guo-Ping Chang-Chien

**Affiliations:** 1Institute of Environmental Toxin and Emerging Contaminant, Cheng Shiu University, Kaohsiung 83347, Taiwan; 0619@gcloud.csu.edu.tw (H.-C.K.); buingochan2505@gmail.com (N.-H.B.); 2Department of Mechanical and Institute of Mechatronic Engineering, Cheng Shiu University, Kaohsiung 83347, Taiwan; huangbw@gcloud.csu.edu.tw; 3Center for Environmental Toxin and Emerging-Contaminant Research, Cheng Shiu University, Kaohsiung 83347, Taiwan

**Keywords:** indolamines, 5-hydroxytryptophan, LC-MS/MS, probiotic strains, serotonin, sustainability, melatonin, yeast strain

## Abstract

Tryptophan metabolites, such as 5-hydroxytryptophan (5-HTP), serotonin, and melatonin, hold significant promise as supplements for managing various mood-related disorders, including depression and insomnia. However, their chemical production via chemical synthesis and phytochemical extraction presents drawbacks, such as the generation of toxic byproducts and low yields. In this study, we explore an alternative approach utilizing *S. cerevisiae* STG S101 for biosynthesis. Through a series of eleven experiments employing different combinations of tryptophan supplementation, Tween 20, and HEPES buffer, we investigated the production of these indolamines. The tryptophan metabolites were analyzed using liquid chromatography with tandem mass spectrometry (LC-MS/MS). Notably, setups replacing peptone in the YPD media with tryptophan (Run 3) and incorporating tryptophan along with 25 mM HEPES buffer (Run 4) demonstrated successful biosynthesis of 5-HTP and serotonin. The highest 5-HTP and serotonin concentrations were 58.9 ± 16.0 mg L^−1^ and 0.0650 ± 0.00211 mg L^−1^, respectively. Melatonin concentrations were undetected in all the setups. These findings underscore the potential of using probiotic yeast strains as a safer and conceivably more cost-effective alternative for indolamine synthesis. The utilization of probiotic strains presents a promising avenue, potentially offering scalability, sustainability, reduced environmental impact, and feasibility for large-scale production.

## 1. Introduction

5-hydroxytryptophan (5-HTP), 5-hydroxytryptamine (serotonin), and melatonin are tryptophan metabolites involved in several physiological processes, including memory, sleep, social behavior, and pain [1]. These indolamines are synthesized from tryptophan, an essential amino acid that is readily available in protein-based foods including nuts, meat, fish, and eggs. However, most animals, including humans, cannot synthesize tryptophan, which they get exclusively from diet. Plants, bacteria, fungi, archaea, and algae synthesize tryptophan via the shikimate pathway, which has not been found in animal cells [2]. 5-hydroxytryptophan is an intermediate in tryptophan metabolism but it is also found in small amounts in protein-based foods. It is a dietary supplement sold over the counter and on the internet in tablet or powder form [3]. Commercially, 5-hydroxytryptophan is extracted from the seeds of the West African shrub *Griffonia simplicifolia*. Similarly, melatonin is one of the most widely prescribed drugs used to treat sleep disorders, with a market value of USD 851 million in 2016 [4]. According to ClinCalc, melatonin was the 298th most prescribed drug in the US (the largest melatonin market) in 2020 with over 1 million prescriptions [5]. Most of the melatonin supplement is chemically synthesized from organic precursors such as 5-methoxy-3-indolylacetonitrile and phthalimide [6,7]. This process results in a yield of ~80% and several unwanted side products, e.g., formaldehyde, that pose health risks [7]. Additionally, some products claim to contain phytomelatonin, which is melatonin extracted from plants such as tart cherry [6].

Probiotics are defined as “live microorganisms that are beneficial to the health of the host when administered in sufficient amounts” by the World Health Organization (WHO) [8]. Probiotics are found in several fermented foods, e.g., cheese, yogurt, kimchi, sauerkraut, and miso. The most common probiotics found in food include *Lactobacillus* spp., *Streptococcus* spp., *Bifidobacterium* spp., and *Saccharomyces* spp. [9,10,11]. Microorganisms also metabolize L-tryptophan into serotonin and melatonin [12,13,14,15]. However, the physiological role of these neurotransmitters and the synthesis pathways in microbes are not well understood [2]. Gonçalves et al. [2] provided a putative metabolic pathway for bacteria whereby tryptophan was converted to serotonin via 5-HTP through phenylalanine hydroxylase and aromatic amino acid decarboxylase activity; thereafter, the serotonin was converted to melatonin via 5-methoxytryptamine or N-acetylserotonin through N-acetylserotonin *o*-methyltransferase and serotonin–N-acetyltransferase enzymes. An alternative pathway highlighted by the study involved the conversion of tryptophan into tryptamine by tryptophan decarboxylase. Ma et al. [14] studied the bacterial melatonin synthesis pathway using an endophytic strain *Pseudomonas fluorescens* RG11 isolated from the roots of Red Globe grape cultivar. They achieved this using ^15^N double-labeled L-tryptophan. They found that the ^15^N-tryptophan was converted into ^15^N-hydroxytryptophan, and that ^15^N-tryptamine was not detected. They used this observation to postulate that hydroxylation catalyzed by a hydroxylase enzyme occurred as the first metabolic step. However, no investigation of the genes was conducted to confirm the results. The results show that the isotopic melatonin and other metabolites increased between 0 and 30 h post-incubation. However, the production peaked at 6 h and then sharply decreased in the growth phase.

Microbial species have been used to study indolamine synthesis, including probiotic species. Özoğul et al. [15] was one of the first studies to show that probiotic strains can synthesize indolamines. In the study, four probiotic strains, *L. lactis* subsp. *cremoris*, *L. lactis* subsp. *lactis*, *L. plantarum*, and *S. thermophilus*, were cultured in arginine decarboxylase broth, and biogenic amine synthesis was monitored. However, only tryptamine and serotonin were monitored, and the media and substrate were not suitable for high yields of the indolamines. In another study, Fernandez-Cruz et al. [16] provided two probiotic yeast strains, *S. cerevisiae* QA23 and P24, with additional 1 mM L-tryptophan to increase the yield of indolamines. As a result, 5-HTP, serotonin, melatonin, and other L-tryptophan metabolites, including tryptophol and L-tryptophan ethyl ester, were all synthesized during the incubation. In most cases, the indolamine concentrations after the addition of tryptophan were higher than without additional tryptophan. For example, the serotonin concentration produced by *S. cerevisiae* P24 was 1037 ± 13.2 pg per 10^9^ cells compared to 681 ± 3.11 pg per 10^9^ cells, respectively. Besides using probiotic species to synthesize the commercially important indolamines, alternative approaches focus on modulating gut microbiota, consequently influencing serotonin levels in the body. BioGaia AB filed a patent application for serotonin-producing lactic acid bacterial strains of *L. reuteri* for use in treating serotonin deficiency [17]. Furthermore, the California Institute of Technology holds a patent for adjusting the composition of gut microbiota to regulate serotonin levels in individuals [18].

Through a series of eleven meticulously designed experiments, this study aims to explore the capacity of an *S. cerevisiae* strain to synthesize tryptophan metabolites, with a focus on 5-HTP, tryptamine, serotonin, and melatonin, due to their commercial value. By employing various combinations of tryptophan supplementation, Tween 20, and HEPES buffer, we aim to assess the strain’s growth success in the media. Furthermore, we will utilize liquid chromatography with tandem mass spectrometry (LC-MS/MS) to analyze sixteen tryptophan metabolites, with particular emphasis on 5-HTP, serotonin, and melatonin. The insights garnered from this research are poised to significantly advance our understanding of tryptophan metabolite biosynthesis within a probiotic yeast strain. Moreover, this investigation represents a crucial stride towards the development of a safe and economically viable alternative for the commercial production of essential indolamines.

## 2. Results

### 2.1. Growth and Biosynthesis of Tryptophan Metabolites by S. cerevisiae STG S101 Grown in YPD Media

#### 2.1.1. Plate Count and Growth Curve

The plate count, as depicted in Table 1, was employed to observe the colony morphology of the yeast strain grown in YPD media. The growth was monitored on plates inoculated with the yeast strain at dilution levels of 10^8^ and 10^9^ and incubated for 3, 4, and 5 days. The colonies were beige in color and generally appeared round and opaque. They possessed a smooth and glossy texture, typically raised or domed in the center, consistent with characteristics observed in *S. cerevisiae* grown in YPD media. Furthermore, the growth curve of *S. cerevisiae* STG S101 in YPD media (Figure 1) was analyzed to establish a baseline for subsequent experimental runs. As illustrated in the figure, the lag phase lasted approximately 1 h, followed by an exponential log phase lasting ~4 h (from two to six hours of incubation). The specific growth rate during this period was determined to be 1.83 h^−1^. Subsequently, the deceleration phase, where the cell division slows down, continued until the 12th hour of incubation. Finally, the stationary phase, characterized by a plateau in cell number, where growth rate equals death rate, persisted until the conclusion of the experiment.

#### 2.1.2. Biosynthesis of Tryptophan Metabolites by *S. cerevisiae* STG S101 Grown in YPD Media

The concentrations of the sixteen tryptophan metabolites synthesized by the yeast strain grown in YPD media are detailed in Appendix A. Measurements were taken at intervals of 0, 4, 8, 12, 16, 20, and 24 h of incubation. Throughout this period, the total concentrations exhibited minimal variation, with the lowest (3.27 mg L^−1^) and highest (3.48 mg L^−1^) concentrations observed at the 8th and 12th hour, respectively. Indole-3-lactic acid emerged as the predominant metabolite, accounting for 71.0–86.8% of the total metabolite concentrations, followed by kynurenine, which accounted for 6.38–15.9%. Serotonin and melatonin were not detectable at any point. However, trace amounts of 5-hydroxytryptophan (5-HTP) and tryptamine were identified, reaching maximum concentrations of 0.09 ± 0.00583 mg L^−1^ and 0.0608 ± 0.00327mg L^−1^, respectively.

### 2.2. Growth and Biosynthesis of Tryptophan Metabolites by S. cerevisiae Grown in YPD Media with 50% of Peptone Replaced by Tryptophan

#### 2.2.1. Plate Count and Growth Curve

Table 2 presents the colony morphology of the yeast strain in YPD media with 50% of peptone replaced by tryptophan. The growth was monitored on Petri dishes inoculated with yeast at 10^8^ and 10^9^ dilution levels and incubated for 3, 4, and 5 days. The color, shape, and texture of the yeast remained unchanged compared to the cultures grown in the original YPD media (Run 1). However, the size of the colonies, particularly noticeable at 10^8^ dilution levels, appeared significantly smaller in comparison. Additionally, the growth phases were less distinct, as depicted in Figure 2. These observations suggest that the growth of the *S. cerevisiae* STG S101 was slightly inhibited by the substitution of 50% peptone with tryptophan.

#### 2.2.2. Biosynthesis of Tryptophan Metabolites by *S. cerevisiae* STG S101 Grown in YPD Media with 50% Peptone Replaced by Tryptophan

The concentration values of the sixteen tryptophan metabolites synthesized by *S. cerevisiae* STG S101 grown in YPD media with 50% peptone replaced by tryptophan are presented in Appendix A. Measurements were taken at intervals of 0, 4, 8, 12, 16, 20, and 24 h of incubation. The total metabolite concentrations increased significantly by a factor of 15.6–18.7 compared to Run 1. The lowest (54.0 mg L^−1^) and highest (63.8 mg L^−1^) total metabolite concentrations were observed at the 0th and 24th hour of incubation, respectively. Similarly, indole-3-lactic acid emerged as the predominant metabolite, accounting for 64.5–79.6% of the total metabolite concentrations, followed by kynurenine (7.13–14.3%). Serotonin and melatonin were also not detectable at any point. However, 5-HTP and tryptamine reached maximum concentrations of 6.17 ± 0.676 mg L^−1^ and 2.44 ± 0.0413 mg L^−1^, respectively. They accounted for 7.57–9.68% and 3.79–4.31% of the total metabolite concentrations. In summary, although the growth of *S. cerevisiae* STG S101 was slightly inhibited by the substitution of 50% peptone with tryptophan, the synthesis of tryptophan metabolites significantly increased.

### 2.3. Growth and Biosynthesis of Tryptophan Metabolites by S. cerevisiae Grown in YPD Media with 100% of Peptone Replaced by Tryptophan

Building on the insights gained from the initial two experimental runs (Section 2.1 and Section 2.2), further experiments were conducted, wherein the peptone in the YPD was entirely substituted with tryptophan. Due to the low solubility of tryptophan in water, Tween 20 (0.1% and 0.2%) and HEPES buffer (25 mM and 100 mM) were supplemented in the media, as shown in Table 3. The selected concentrations of Tween 20 and HEPES buffer were based on the results of two preliminary Taguchi-designed tests using 0%, 0.05%, 0.1%, 0.15%, and 0.2% of Tween 20 and 0 mM, 10 mM, 15 mM, 20 mM, 25 mM, 75 mM, and 100 mM of HEPES buffer. The growth curves were then analyzed to assess the influence of the media composition, surfactant, and buffer on the growth of *S. cerevisiae* STG S101. Additionally, the tryptophan metabolites were compared, with a particular focus on 5-HTP, tryptamine, serotonin, and melatonin. Tryptamine was also included due to its role as a precursor to serotonin and melatonin.

#### 2.3.1. Growth Curve

The growth curves of the experimental runs involving *S. cerevisiae* STG S101 grown in YPD media, wherein peptone was entirely substituted by tryptophan, are illustrated in Appendix A. Remarkably, complete replacement of peptone in YPD with tryptophan yielded a more pronounced curve with distinct growth phases compared to the 50% replacement scenario (Appendix A). The doubling time was approximately 4 h, with a specific growth rate of 0.886 h^−1^. Notably, discernible growth phases were also evident in Appendix A,j, corresponding to experimental runs 4, 6, and 10, as detailed in Table 3. The growth of the yeast was most hindered in the experimental runs utilizing 100 mM of HEPES buffer, while the run employing only 0.1% Tween 20 exhibited the most favorable growth. In summary, although complete replacement of peptone with tryptophan slightly impeded the growth of *S. cerevisiae* STG S101, the impact was not overly detrimental.

#### 2.3.2. Biosynthesis of Tryptophan Metabolites by *S. cerevisiae* STG S101 Grown in YPD with 100% of the Peptone Replaced by Tryptophan

The analysis of the sixteen tryptophan metabolites yielded intriguing results (Appendix A). Indole-3-lactic acid remained the predominant metabolite among all the runs, comprising over 98% of the total metabolite concentrations, except for runs 3 and 4, where 5-HTP, 3-hydroxyanthranilic acid, tryptophol, and tryptamine were the primary metabolites. In these instances, these metabolites collectively accounted for 83.1–88.8% and 85.0–92.5% of the total metabolites, respectively. Additionally, serotonin production was only observed in the two experimental runs. These findings indicate that the composition of the media, surfactant, and buffer influenced the metabolism of the yeast strain. In our investigation, we specifically focused on experimental runs 3 and 4 to examine the temporal dynamics of metabolite concentrations using the repeated measured ANOVA analysis. We aimed to discern any notable differences in mean concentrations over time for each metabolite and to identify underlying patterns among the concentrations measured at different periods (Appendix A). For both runs, the Mauchly’s tests were significant (*p* < 0.05), indicating a violation of the sphericity, i.e., the variances of the differences between all possible pairs of conditions are equal. Therefore, there were significant differences in metabolite concentration across the seven testing periods (0, 4, 8, 16, 20, and 24 h). Further support for these findings came from the F-statistic and associated *p*-value of the tests of within-subject effects, which also indicate significant differences in mean concentrations across periods.

Figure 3 and Figure 4 depict serotonin, 5-HTP, and tryptamine production by *S. cerevisiae* STG S101 in the eleven experimental setups. Runs 3 and 4 exhibited the highest concentrations of 5-HTP and tryptamine compared to the other setups. In run 3, the concentrations ranged from 42.3 ± 2.47 to 51.5 ± 4.94 mg L^−1^ for 5-HTP and 12.5 ± 0.768 mg L^−1^ to 16.1 ± 2.44 mg L^−1^ for tryptamine. In run 4, the concentrations ranged from 48.3 ± 3.98 mg L^−1^ to 58.9 ± 16.0 mg L^−1^ for 5-HTP and 11.8 ± 0.39 mg L^−1^ to 18.2 ± 1.04 mg L^−1^ for tryptamine. The serotonin concentrations ranged from 0.0585 ± 0.00092 mg L^−1^ to 0.0642 ± 0.00426 mg L^−1^ and 0.0598 ± 0.0062 mg L^−1^ to 0.0650 ± 0.00211 mg L^−1^, respectively.

## 3. Discussion

The results obtained from the experimental runs investigating the substitution of peptone with tryptophan in YPD media reveal significant insights into the growth dynamics and metabolic responses of *S. cerevisiae* STG S101. Notably, complete replacement of peptone with tryptophan led to more pronounced growth curves with distinct phases compared to partial replacement scenarios. Despite a slight hindrance in growth, indicated by a slightly increased doubling time and specific growth rate, discernible growth phases were evident among the various experimental setups. Interestingly, the presence of different concentrations of HEPES buffer and Tween 20 influenced yeast growth, with high concentrations of HEPES buffer demonstrating the least favorable growth conditions.

The findings suggest that substituting 50% of peptone with tryptophan in YPD media influences the synthesis of tryptophan metabolites by *S. cerevisiae* STG S101. Despite a slight inhibition in yeast growth, the overall production of tryptophan metabolites notably increased. In place of peptone, tryptophan was expected to serve as a nitrogen source. However, it not only served that purpose, but the yeast also metabolized it into its numerous metabolites. This implies that tryptophan availability plays a crucial role in regulating the metabolic pathways leading to the synthesis of these metabolites. The significant elevation in total metabolite concentrations, particularly with indole-3-lactic acid being the predominant metabolite, underscores the metabolic adaptation of the yeast to the altered nutrient environment. Furthermore, the analysis of tryptophan metabolites highlighted the metabolic versatility of *S. cerevisiae* STG S101 in response to changes in media composition. While indole-3-lactic acid remained the predominant metabolite in most runs, the experimental setups with complete substitution of peptone showed a shift towards other metabolites, such as 5-HTP, 3-hydroxyanthranilic acid, tryptophol, and tryptamine. This suggests that variations in media composition, including the presence of surfactants and buffers, can influence the metabolic pathways involved in tryptophan metabolism. However, the exact influence of each variable would require further investigation.

Experimental runs 3 and 4 had the highest 5-HTP and tryptamine concentrations. Furthermore, only these two runs had serotonin concentrations. Table 4 compares the highest concentrations of 5-HTP, tryptamine, serotonin, and melatonin in our study with those from previous studies. Notably, the serotonin concentration values were higher than those reported by Ma et al. [14] using a bacterial strain, *Pseudomonas fluorescens* RG11 (0.008 ± 0.00065 mg L^−1^). However, Özoğul et al. [15], employing probiotic bacterial strains, reported much higher serotonin concentrations: 0.71 ± 0.08 mg L^−1^ (*L. lactis* subsp. *cremoris*), 0.70 ± 0.05 mg L^−1^ (*L. lactis* subsp. *lactis*), 0.91 ± 0.07 mg L^−1^ (*L. plantarum*), and 2.70 ± 0.06 mg L^−1^ (*S. thermophilus*). Fernandez-Cruz et al. [16] reported melatonin synthesis by one strain of *S. cerevisiae*. This comparison highlights the variability in serotonin and melatonin production across different microbial species and strains and the influence of experimental conditions on metabolite yield.

The detection of 5-HTP, tryptamine, and serotonin at appreciable concentrations suggests potential applications in biotechnological processes or pharmaceutical production, where these compounds are of interest. Notably, recent patent filings show an interest in this research. For example, Grasset et al. [17] identified serotonin-producing probiotic bacterial strains, and Hsiao and Yano [18] regulated serotonin levels in subjects by modulating their gut microbes. This study contributes to this growing research interest in using probiotic species to produce valuable tryptophan metabolites.

Overall, these findings provide valuable insights into the metabolic responses of *S. cerevisiae* to alterations in nutrient availability and highlight potential avenues for optimizing the production of bioactive compounds in yeast-based bioprocessing applications. Overall, these findings provide insights into the metabolic responses of *S. cerevisiae* to changes in nutrient composition, offering avenues for further exploration in both basic research and applied contexts, such as bioprocessing and drug development.

## 4. Materials and Methods

### 4.1. Chemicals

Methanol (CH_3_OH), ammonium acetate (CH_3_COONH_4_), acetonitrile (CH_3_CN), picolinic acid, 3-hydroxykynurenine, quinolinic acid, serotonin, 5-hydroxytryptophan, kynurenine, 3-hydroxyanthranilic acid, tryptamine, L-tryptophan, 5-hydroxyindole, acetic acid, indoxyl sulfate, N-Acetylserotonin, xanthurenic acid, indole-3-acetamide, kynurenic acid, DL-Indole-3-lactic acid, indole-3-carboxaldehyde, indole-3-acetic acid, tryptophol, melatonin, 5-hydroxyindole acetic acid-D_5_, serotonin-D_4_, indole-3-lactic acid-D_4_, kynurenic acid-D_5_, melatonin-D_4_, picolinic acid-D_3_, tryptamine-D_4_, xanthurenic acid-D_4_, 3-hydroxyanthranilic acid-D_3_, 3-hydroxykynurenine-^13^C_2_-^15^N, 5-Hydroxytryptophan-D_4_, indole-3-acetamide-D_5_, kynurenine-D_4_, L-Tryptophan-^13^C_11_,^15^N, and tryptophan-D_5_ were purchased from Sigma Aldrich, Merck KGaA (Darmstadt, Germany) and other reagents and chemicals used in the study were purchased from local suppliers. The reagents and chemicals were of analytical grade or higher purity. Additionally, the yeast strain *S. cerevisiae* STG S101 was purchased from Fermentis (Marcq-En-Baroeul, France).

### 4.2. Experimental Design

A series of eleven experiments were designed to investigate the synthesis of sixteen tryptophan metabolites, as detailed in Table 3. In the study, peptone in the YPD media was gradually substituted with tryptophan (0%, 50%, and 100%). Given that peptone serves as a nitrogen source, the required amount of tryptophan was adjusted based on the total nitrogen content, considering peptone with 15.1% and tryptophan with 13.7% total nitrogen. Due to tryptophan’s low solubility in water (11.4 g L^−1^ at 25 °C), a combination of Tween 20 and HEPES buffer was employed to improve its solubility. A tryptophan solubility test was carried out prior to the experiment to determine the respective concentrations of Tween 20 and HEPES buffer. The Taguchi design was utilized to devise setups encompassing 100% tryptophan, varying concentrations of Tween 20 (0%, 0.1% and 0.2%), and HEPES buffer (0 mM, 25 mM, and 100 mM). Incubation experiments were carried out at 30 °C, the optimal temperature for *S. cerevisiae*. The procedures for preparing the culture media, inoculum, and incubation followed the steps presented by Olivares-Marin et al. [19]. The plate count agar method to evaluate the colony morphology and the effect of different growth media followed the Taiwan Food and Drug Administration method MOHWM0008.01 on Methods of Test for Food Microorganisms—Test of Mold and Yeast Count. All the experiments were carried out in triplicate.

### 4.3. Probiotic Incubation and Indolamine Biosynthesis Procedures

#### 4.3.1. Culture Media and Inoculum Preparation

First, 100 mL of 2% yeast–peptone–dextrose (YPD) liquid medium was prepared by adding 1 g of yeast extract, 2 g of peptone, and 2 g of glucose to 100 mL of distilled water. For experiments requiring peptone replacement with tryptophan, the amount of tryptophan required was corrected to the total nitrogen. Then, 4.8 mL of the media was dispensed into sterilizable 15 mL conical tubes, followed by autoclaving the media for 15 min at 121 °C and 14.7 psi. Next, 1 g of the yeast strain was transferred into a 9 mL test tube containing the 2% YPD liquid culture medium and incubated overnight at 30 °C with agitation at 200 rpm. The cells grown in the conical tube were streaked onto a Petri dish (60 × 15 mm) filled with sterile 2% YPD agar, prepared by adding 10 g of yeast extract, 20 g of peptone, 20 g of glucose, and 20 g of bacteriological agar to 1000 mL of distilled water. The Petri dish was incubated at 30 °C until isolated colonies showed growth. Afterwards, the pre-inoculum was prepared by inoculating 1 isolated colony into a conical tube containing 10 mL of cool, sterile 2% YPD broth and incubated overnight at 30 °C with continuous agitation at 200 rpm.

#### 4.3.2. Growth Curve Study

A 20 mL flask was filled with 4.8 mL of the 2% YPD liquid culture medium and sterilized. Then, 0.2 mL of the prepared pre-inoculum presented in Section 4.3.1 was added and incubated at 250 rpm and 30 °C for 24 h. During the initial 12-h period, samples of the culture solution were collected every 1 h, each consisting of 0.1 mL, and diluted with 0.9 mL of distilled water. The optical density (OD) was subsequently measured for each sample at 600 nm. For the subsequent 12 h, samples of the culture solution were collected every 4 h, again consisting of 0.1 mL, and diluted with 0.9 mL of distilled water. Similarly, the OD at 600 nm was then measured for these samples. Finally, the test results were multiplied by 10 to yield the final readings.

### 4.4. Sample Preparation and Analysis

Standard stock solutions (1000 mg L^−1^) of the twenty analytes were individually prepared in methanol. A mixed standard solution containing the analytes at a concentration of 1000 μg L^−1^ was generated by obtaining aliquots of each individual stock solution. These solutions were stored in a freezer at −20 °C prior to use. The mixed standard solution was appropriately diluted with methanol to prepare the working solution series. Additionally, a mixed internal standard solution in methanol, with a concentration of 200 μg L^−1^, was prepared using the mixed standard solution.

Sixteen tryptophan metabolites were analyzed in samples collected after every four hours of incubation (0, 4, 8, 12, 16, 20, and 24 h). Prior to the analysis of the metabolites, the samples were first thawed to room temperature and thoroughly mixed before transferring 200 μL into 1.5 mL centrifuge tubes. Following this, a mixture consisting of 400 μL of chilled internal standard working solution, along with 200 μL of acetonitrile and 400 μL of methanol, was added to each tube. After thorough mixing, the mixture underwent centrifugation at 500 rpm for 30 min at 4 °C. To enhance protein precipitation, the centrifuged mixture was then placed at −20 °C for 1 h under light protection. Subsequently, the sample was centrifuged at 14,000 rpm for 15 min at 4 °C, and the resulting supernatant was carefully collected and concentrated to 100 μL through centrifugation. Following concentration, a mixture of 100 μL of a 5 mM ammonium acetate aqueous solution with methanol (9/1) was added, thoroughly mixed, and then injected into a high-performance liquid chromatography (HPLC) with a tandem mass spectrometer (MS/MS) for analysis. The concentration values of the metabolites were represented as mean concentrations with standard deviations.

The LC instrument setup consisted of an Agilent 1260 Infinity II HPLC and an Agilent 6470A Triple Quadrupole MS/MS. Chromatographic separation was performed with a Water Acquity UPLC high-strength silica (HSS) T3 1.8 µm (2.1 mm × 100 mm) column (Waters, Wilmslow, UK). The mass transition ions, fragmentor voltages, and collision energies of the analytes are presented in Appendix A. The sixteen tryptophan metabolites were eluted by a gradient of 5 mM ammonium acetate aqueous solution (mobile phase A) and acetonitrile (mobile phase B). The gradient conditions of the mobile phase were as follows: 0–0.5 min, 0% B; 0.5–2.5 min, 0–10% B; 2.5–3.5 min, 10–15% B; 3.5–4.5 min, 15–25% B; 4.5–5.5 min, 25–35% B; 5.5–6.5 min, 35–45% B; 6.5–7.0 min, 45–55% B; 7.0–7.5 min, 55–100% B; 7.5–10.5 min, 100% B; 10.5–14.5 min, 100-0% B; and 14.5–25 min, 0% B. The flow rate was 0.3 mL min^−1^, and the column oven was set to 40 °C. The autosampler was maintained at 4 °C throughout the analyses. Positive and negative ionization modes for electrospray ionization source were used for detection. The source conditions were as follows: gas temperature of 350 °C, flow rate of 9 L min^−1^, and capillary voltage of 4000 V(+) and 2500 V(−). Multiple reaction monitoring (MRM) mode was used. The data were acquired and processed using Agilent Mass Hunter Workstation Version 9.0 software. The results of the mass spectrometer demonstrated that the R-squared of the 20 analytes were no less than 0.990. The linear range for the 20 analytes was 5~200 μg L^−1^.

### 4.5. Statistical Analysis

The temporal dynamics of metabolite concentration were investigated using repeated measured analyses of variance (ANOVA) performed by SPSS.

## Figures and Tables

**Figure 1 ijms-25-04747-f001:**
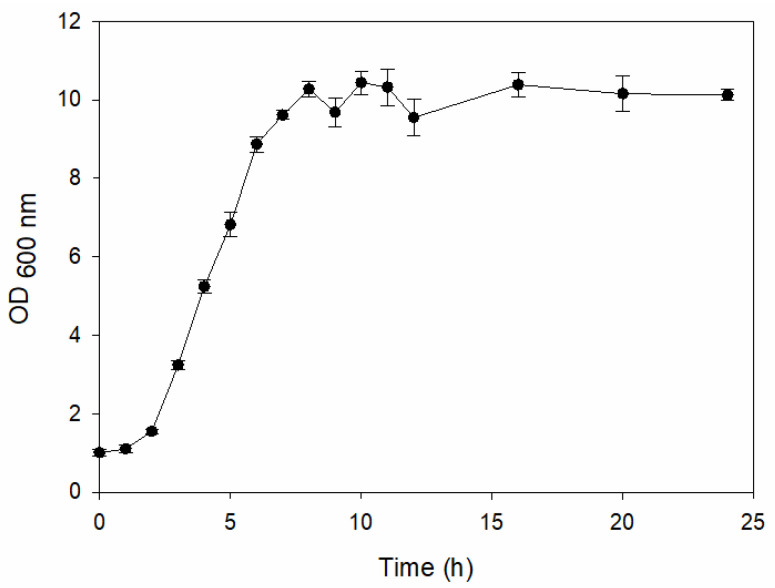
The growth curve of S. cerevisiae STG S101 grown in YPD media.

**Figure 2 ijms-25-04747-f002:**
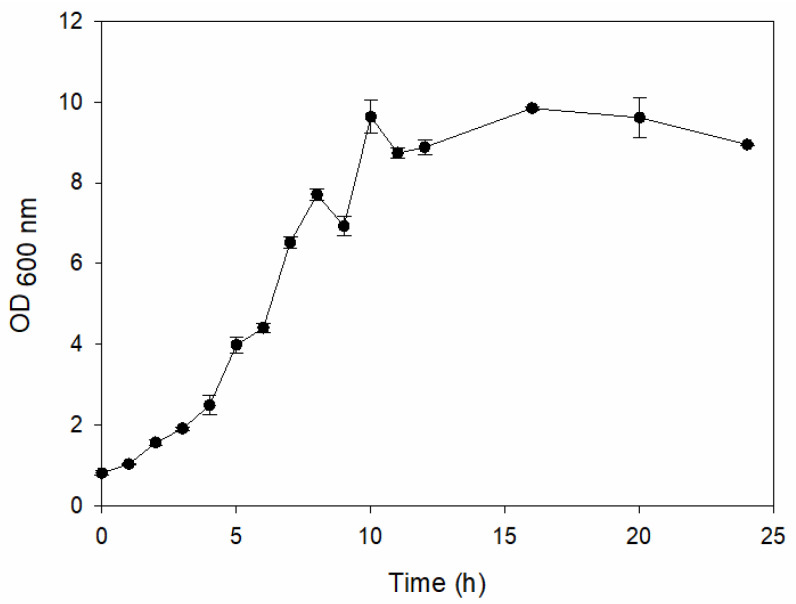
Growth of *S. cerevisiae* STG S101 in YPD media with 50% peptone replaced by tryptophan.

**Figure 3 ijms-25-04747-f003:**
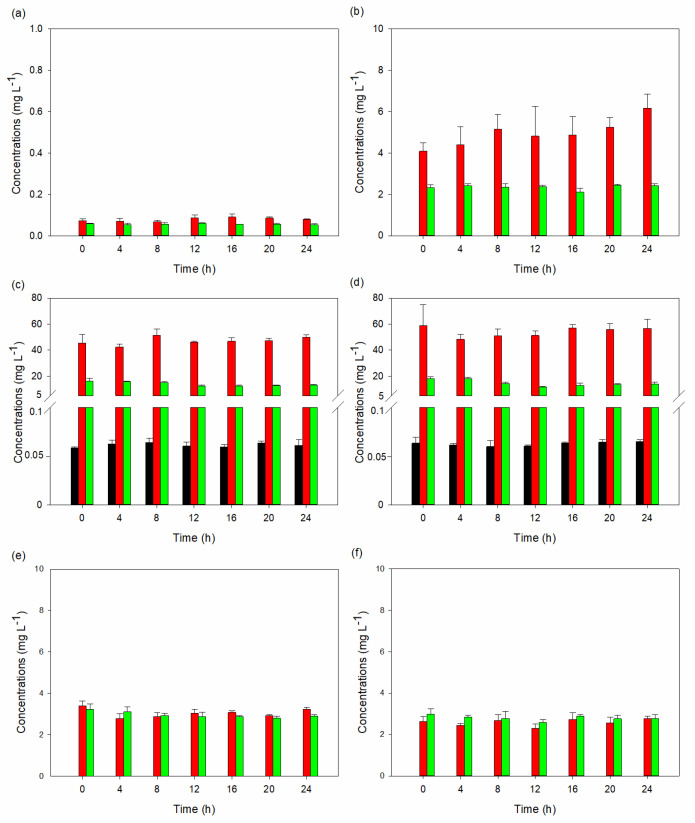
Average concentrations with standard deviation of serotonin, 5-hydroxytryptophan, and tryptamine synthesis by *S. cerevisiae* STG 101 in various growth conditions: (**a**) normal YPD media; (**b**) YPD media with 50% peptone replaced by tryptophan; (**c**) YPD media with 100% tryptophan; (**d**) YPD media with 100% tryptophan and 25 mM HEPES buffer; (**e**) YPD media with 100% tryptophan and 100 mM HEPES buffer; and (**f**) YPD media with 100% tryptophan and 0.1% Tween 20. Serotonin, 5-hydroxytryptophan, and tryptamine are represented by black, red, and green colors, respectively. Serotonin was only detected in (**c**,**d**).

**Figure 4 ijms-25-04747-f004:**
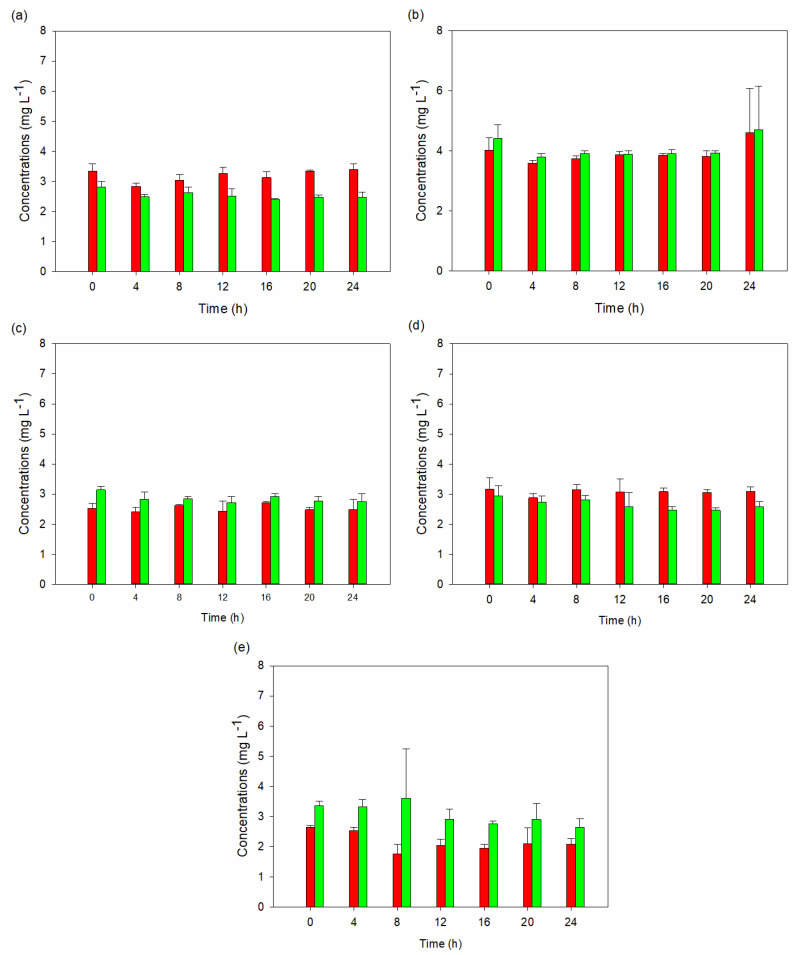
Average concentrations with standard deviation of serotonin, 5-hydroxytryptophan, and tryptamine synthesis by *S. cerevisiae* STG 101 in various growth conditions: (**a**) YPD media with 100% tryptophan, 0.1% Tween 20, and 25 mM HEPES buffer; (**b**) YPD media with 100% tryptophan, 0.1% Tween 20, and 100 mM HEPES buffer; (**c**) YPD media with 100% tryptophan and 0.2% Tween 20; (**d**) YPD media with 100% tryptophan, 0.2% Tween 20, and 25 mM HEPES buffer; and (**e**) YPD media with 100% tryptophan, 0.2% Tween 20, and 100 mM HEPES buffer. 5-hydroxytryptophan and tryptamine are represented by red and green colors, respectively. Serotonin was not detected in any of the conditions.

**Table 1 ijms-25-04747-t001:** Plate count of *S. cerevisiae* STG S101 in YPD media at 10^8^ and 10^9^ dilution levels after 3, 4, and 5 days of incubation.

	Day 3	Day 4	Day 5
10^8^	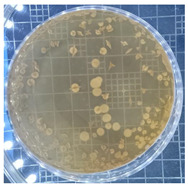	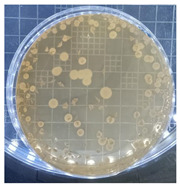	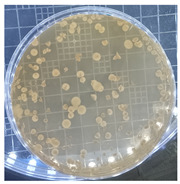
10^9^	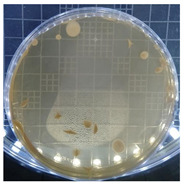	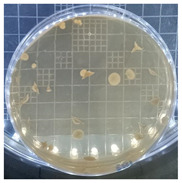	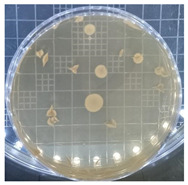

**Table 2 ijms-25-04747-t002:** Plate count of *S. cerevisiae* STG S101 in YPD media with 50% peptone replaced by tryptophan at 10^8^ and 10^9^ dilution levels after 3, 4, and 5 days of incubation.

	Day 3	Day 4	Day 5
10^8^	* 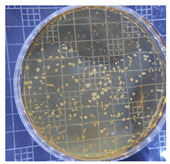 *	* 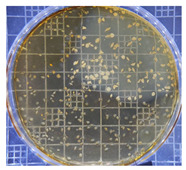 *	* 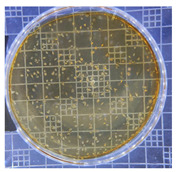 *
10^9^	* 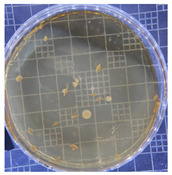 *	* 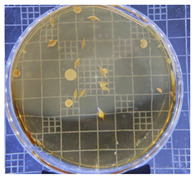 *	* 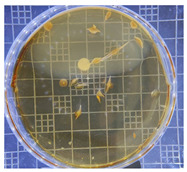 *

**Table 3 ijms-25-04747-t003:** Summary of experiments conducted with different combinations of peptone and tryptophan in the YPD medium, along with varying concentrations of Tween 20 and HEPES buffer.

Runs	Peptone: Tryptophan	Tween 20 (%)	HEPES Buffer (mM)
1	100%:0%	0	0
2	50%:50%	0	0
3	0%:100%	0	0
4		0	25
5		0	100
6		0.1	0
7		0.1	25
8		0.1	100
9		0.2	0
10		0.2	25
11		0.2	100

**Table 4 ijms-25-04747-t004:** Comparison of microbial synthesis of target tryptophan metabolites from the literature.

Microbial Species/Strain	Source	Substrate	Concentrations	Ref.
*S. cerevisiae* STG S101	Purchased	YPD medium (with 100% tryptophan and 25 mM HEPES buffer)	5-HTP = 58.9 ± 16.0 mg L^−1^ Tryptamine = 18.2 ± 1.04 mg L^−1^ Serotonin = 0.0643 ± 0.00272 mg L^−1^	This study
*L. lactis* subsp. *cremoris* (MG 1363)	Purchased	Arginine decarboxylase broth (ADB)	Tryptamine = 0.00 Serotonin = 0.71 ± 0.08 mg L^−1^	[15]
*L. lactis* subsp. *lactis* (IL 1403)			Tryptamine = 0.00 Serotonin = 0.70 ± 0.05 mg L^−1^	
*L. plantarum* (FI8595)			Tryptamine = 0.00 Serotonin = 0.91 ± 0.07 mg L^−1^ broth	
*S. thermophilus* (NCFB2392)			Tryptamine = 19.6 ± 0.71 mg L^−1^ broth Serotonin = 2.70 ± 0.06 mg L^−1^ broth	
*S. cerevisiae* QA23	Purchased	Synthetic must (200 g L^−1^ reducing sugars + 300 mg N L^−1^ assimilable nitrogen), more L-tryptophan (1 mM)	5-HTP = 949 ± 80.3 pg/10^9^ Serotonin = 1308 ± 147 pg per 10^9^ cells Melatonin = ND	[16]
*S. cerevisiae* P24			5-HTP = 935 ± 25.2 pg/10^9^ Serotonin = 1037 ± 13.2 pg per 10^9^ cells Melatonin = 60.83 ± 13.44 pg 10^9^ cells	
*P. fluorescens* RG11	Isolated from the roots of Red Globe grape cultivar	200 mg L^−1^ of ^15^N double-labeled L-tryptophan	^15^N-5-HTP = 0.0181 ± 0.00114 mg L^−1^ ^15^N-Serotonin = 0.00828 ± 0.00065 mg L^−1^ ^15^N-acetylserotonin = 0.00866 ± 0.00082 mg L^−1^ ^15^N-melatonin = 0.00132 ± 0.00012 mg L^−1^	[14]

## Data Availability

Data are contained within the article or Appendix A.

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
