# Peer review of "Biosynthetic Pathways of Tryptophan Metabolites in Saccharomyces cerevisiae Strain: Insights and Implications"

_ijms, 2024, doi:10.3390/ijms25094747_

Round 1
Reviewer 1 Report
Comments and Suggestions for Authors
Dear editor and authors:
This article elucidates the profound impact of alterations in the composition of medium on the growth and tryptophan metabolism pathways of S. cerevisiae. In addition, the successful biosynthesis of 5-HTP and serotonin were achieved by optimizing the buffer and surfactant conditions. Generally, the topic is interesting and the study has been well performed. However, a number of technical and language issues should be addressed:
Major issues:
1. It is essential to highlight the correlation between tryptophan metabolites and incorporate relevant introductions to metabolic pathways, thereby enhancing the logical coherence in determining metabolite levels.
2. Insufficient data support exists regarding the selection of Tween 20 and HEPES buffer concentrations and their influence on promoting tryptophan solubility.
3. Considering the structure of manuscript, it is recommended that the reports on microbial synthesis of tryptophan metabolites mentioned in the literature can be excluded from the results or relocated to other sections.
Minor issues:
1. line 112, “concentration” should be “concentrations”
2. line 236,“the the”should be “the”
3. line 316,“4.2.1”should be “4.3.1”
4. line 331,“Internal”should be “internal”
5. line 358,“Negative”should be “negative”
Comments on the Quality of English LanguageDear editor and authors:
This article elucidates the profound impact of alterations in the composition of medium on the growth and tryptophan metabolism pathways of S. cerevisiae. In addition, the successful biosynthesis of 5-HTP and serotonin were achieved by optimizing the buffer and surfactant conditions. Generally, the topic is interesting and the study has been well performed. However, a number of technical and language issues should be addressed:
Major issues:
1. It is essential to highlight the correlation between tryptophan metabolites and incorporate relevant introductions to metabolic pathways, thereby enhancing the logical coherence in determining metabolite levels.
2. Insufficient data support exists regarding the selection of Tween 20 and HEPES buffer concentrations and their influence on promoting tryptophan solubility.
3. Considering the structure of manuscript, it is recommended that the reports on microbial synthesis of tryptophan metabolites mentioned in the literature can be excluded from the results or relocated to other sections.
Minor issues:
1. line 112, “concentration” should be “concentrations”
2. line 236,“the the”should be “the”
3. line 316,“4.2.1”should be “4.3.1”
4. line 331,“Internal”should be “internal”
5. line 358,“Negative”should be “negative”
Author Response
Reviewer 1
This article elucidates the profound impact of alterations in the composition of medium on the growth and tryptophan metabolism pathways of S. cerevisiae. In addition, the successful biosynthesis of 5-HTP and serotonin were achieved by optimizing the buffer and surfactant conditions. Generally, the topic is interesting and the study has been well performed. However, a number of technical and language issues should be addressed:
|
Reply: |
Major issues:
- It is essential to highlight the correlation between tryptophan metabolites and incorporate relevant introductions to metabolic pathways, thereby enhancing the logical coherence in determining metabolite levels.
|
Reply: Thank you very much for the recommendations. We have added relevant information about microbial metabolism of tryptophan in the introduction section. In addition, we performed statistical analysis for the two setups that had serotonin. We performed repeated measured ANOVA to measure if there were significant differences in mean concentrations over time for each metabolite. The added information about the pathway is as follows:
“In our investigation, we specifically focused on experimental runs 3 and 4 to examine the temporal dynamics of metabolite concentrations using the repeated measured ANOVA analysis. We aimed to discern any notable differences in mean concentrations over time for each metabolite and to identify underlying patterns among the concentrations measured at different periods. For both runs, the Mauchly’s tests were significant (p < 0.05) indicating a violation of the sphericity, i.e., the variances of the differences between all possible pairs of conditions are equal. Therefore, there were significant differences in metabolite concentration across the seven testing periods (0, 4, 8, 16, 20, and 24 hours). Further support for these findings came from the F-statistic and associated p-value of the tests of within-subject effects, which also indicate significant differences in mean concentrations across periods.” (Please see lines 218-228) |
- Insufficient data support exists regarding the selection of Tween 20 and HEPES buffer concentrations and their influence on promoting tryptophan solubility.
|
Reply: Thank you for the insightful question. Before selecting the concentrations of Tween 20 and HEPES buffer for our study, we performed preliminary tryptophan solubility experiments. We developed a series of Taguchi experiments to determine the effect of the concentrations. We used Tween 20 concentrations of 0%, 0.05%, 0.1%, 0.15%, and 0.2%, and HEPES buffer concentrations of 0 mM, 10 mM, 15 mM, 20 mM, and 25 mM. We replaced the peptone in 2% YPD media with 2.2 g of tryptophan and added Tween 20 and HEPES buffer concentrations based on the Taguchi design. In total, we had 25 experimental runs. The results were as follows: Preliminary experiment 1: Results of the solubility tests using different concentrations of Tween 20 and HEPES buffer
Preliminary experiment 2: Results of the solubility tests using different concentrations of Tween 20 and HEPES buffer
We analyzed the results and found that the effect of Tween 20 did not change significantly after 0.05%, but HEPES buffer effect appeared to increase with concentrations. We repeated the same experiments with higher concentrations of HEPES buffer (0 mM, 25 mM, 50 mM, 75 mM, and 100 mM) and Tween 20 concentrations of 0%, 0.05%, 0.10%, 0.15% and 0.20% The results from the preliminary experiments informed us on the choice of concentrations for our study. This information has been included in the main text (Please see lines 184-187)
|
- Considering the structure of manuscript, it is recommended that the reports on microbial synthesis of tryptophan metabolites mentioned in the literature can be excluded from the results or relocated to other sections.
|
Reply: |
Minor issues:
- line 112, “concentration” should be “concentrations”
- line 236,“the the”should be “the”
- line 316,“4.2.1”should be “4.3.1”
- line 331,“Internal”should be “internal”
- line 358,“Negative”should be “negative”
|
Reply: We want to thank the reviewer for spotting these errors. We have corrected them accordingly. Furthermore, we proofread our revised manuscript carefully to ensure that there were no further errors. |

Reviewer 2 Report
Comments and Suggestions for Authors
The article submitted for revision refers to the very interesting subject of tryptophan metabolites biosynthesis. Overall, the study is well-designed and conducted. However, there are a few issues that need to be addressed.
Major remarks:
It is not clear what is presented on the charts (Figures 3 and 4). Are these means with standard deviation? If so, from how many repetitions? The information regarding the type of variable depicted on the chart should be included in the chart's caption, while details about the calculation method of the means and the number of repetitions should be provided in the methodology section.
Additionally, the scale on the y-axis is different on each chart, making it difficult to compare various growth conditions. There is also a lack of scale for serotonin on charts c and d. I suggest standardizing the values on the y-axis where possible to a maximum value of 1 (chart a), 10 (charts b and e-k), and 80 (charts c and d).
Furthermore, serotonin is not depicted on charts g-k (Figure 4), despite being mentioned in the charts description. Since it is already stated in the results section (Line 190-191) that serotonin production occurred only in two experimental runs (Figure 3), there is no need to include it in the chart descriptions for Figure 4.
A major flaw of this article is also the absence of a thorough discussion section. Some fragments of discussion appear within the results section (e. g. Lines 200-205). There is also Table 4 (Line 226) which is neither referenced nor discussed anywhere in the text. I understand that with such an research design, it's challenging to completely separate the results from the discussion and conclusions, hence it might be better to combine the discussion with the results. Additionally, the discussion provided in the manuscript is too brief and does not fully address the topic.
Minor remarks:
Lines 200 – 205 If there are separate sections for Results and Discussion, this information should be placed in the Discussion section rather than in the Results section.
Lines 159 – 161: In my opinion it is better to refer the Table 3 after this sentence.
Line 236: delete the double "the"
Line 331: The internal standard should be written in lower case
Line 358: The negative ionization should be written in lower case
Supplementary material:
Figure S1 needs a detailed legend.
Author Response
Reviewer 2
|
We would like to thank the reviewer for the time they spent on peer reviewing our manuscript and giving us valuable comments and suggestions that have helped us improve our manuscript tremendously. We have responded to all of the comments below and made the respective changes in the revised manuscript. The changes in the manuscript are highlighted in red. |
Major remarks:
- It is not clear what is presented on the charts (Figures 3 and 4). Are these means with standard deviation? If so, from how many repetitions? The information regarding the type of variable depicted on the chart should be included in the chart's caption, while details about the calculation method of the means and the number of repetitions should be provided in the methodology section.
|
Reply: Thank you for your insightful question. Figures 3 and 4 depict the mean concentration with standard deviation (n = 3) of serotonin and its precursors 5-hydroxytryptophan and tryptamine found in experimental runs. These metabolites are of commercial interest and that is why we focused on depicting them and not all metabolites. We have included the concentration values of the other metabolites in supplementary files. The information about the variable depicted in the figures, calculation method of the means, and number of repetitions has been included in the figure captions and methodology section, as marked in red in the main text.
|
- Additionally, the scale on the y-axis is different on each chart, making it difficult to compare various growth conditions. There is also a lack of scale for serotonin on charts c and d. I suggest standardizing the values on the y-axis where possible to a maximum value of 1 (chart a), 10 (charts b and e-k), and 80 (charts c and d).
|
Reply: Thank you for this useful recommendation. We have revised the figures accordingly. |
- Furthermore, serotonin is not depicted on charts g-k (Figure 4), despite being mentioned in the charts description. Since it is already stated in the results section (Line 190-191) that serotonin production occurred only in two experimental runs (Figure 3), there is no need to include it in the chart descriptions for Figure 4.
|
Reply: |
- A major flaw of this article is also the absence of a thorough discussion section. Some fragments of discussion appear within the results section (e. g. Lines 200-205). There is also Table 4 (Line 226) which is neither referenced nor discussed anywhere in the text. I understand that with such an research design, it's challenging to completely separate the results from the discussion and conclusions, hence it might be better to combine the discussion with the results. Additionally, the discussion provided in the manuscript is too brief and does not fully address the topic.
|
Reply: |
Minor remarks:
- Lines 200 – 205 If there are separate sections for Results and Discussion, this information should be placed in the Discussion section rather than in the Results section.
|
Reply: |
- Lines 159 – 161: In my opinion it is better to refer the Table 3 after this sentence.
|
Reply: |
- Line 236: delete the double "the"
- Line 331: The internal standard should be written in lower case
- Line 358: The negative ionization should be written in lower case
- Supplementary material:
- Figure S1 needs a detailed legend.
|
Reply: |

Reviewer 3 Report
Comments and Suggestions for Authors
Overall, this is a repetition of experiments relative to other juń published studies. There is a lack of showing statistical significance. For the purposes of this study, the authors set out to analyze sixteen tryptophan metabolites, we could see only four, what about the rest?
Author Response
Reviewer 3
|
Reply: |
Overall, this is a repetition of experiments relative to other juń published studies. There is a lack of showing statistical significance. For the purposes of this study, the authors set out to analyze sixteen tryptophan metabolites, we could see only four, what about the rest?
|
Reply: |
Round 2
Reviewer 1 Report
Comments and Suggestions for Authors
The current revised version can be accepted.
Reviewer 2 Report
Comments and Suggestions for Authors
I am satisfied with the corrections introduced by the Authors. I only have two minor remarks:
Line 290: change "only this two runs had serotonin concentrations" to "serotonin production was detected solely in these two runs."
Line 292: either "value was" or "values were"
The overall quality of the English language is good, with only occasional errors.